# Autologous Breast Reconstruction with Free Nipple–Areola Graft after Circumareolar (Skin Reducing) Mastectomy

**DOI:** 10.3390/jpm12101588

**Published:** 2022-09-26

**Authors:** Hisham Fansa, Sora Linder

**Affiliations:** 1Department of Plastic Surgery and Breast Center Zürich, Spital Zollikerberg, Zollikerberg, 8125 Zürich, Switzerland; 2Department of Plastic, Reconstructive, and Aesthetic Surgery, Hand Surgery, Klinikum Bielefeld, OWL-University, 33604 Bielefeld, Germany

**Keywords:** immediate autologous breast reconstruction, DIEP-flap, inner thigh-flap, free nipple areola graft, large/ptotic breasts, circumareolar mastectomy, risk-reducing mastectomy, scarless mastopexy, oncological safety

## Abstract

Introduction of skin-sparing mastectomy (SSM) led to a paradigm shift in breast reconstruction. Primary reconstructions have become the therapy of choice. At the same time, immediate autologous reconstructions are oncologically safe and aesthetically pleasing. Our preferred SSM incision is the circumareolar with removal of nipple and areola (NAC). Adjustment of the skin envelope is well accomplished in mild-to-moderate ptotic breasts. We describe our technique consisting of circumareolar incision in SSM, keeping the NAC as a free graft, and immediate autologous reconstruction and immediate free NAC grafting on the flap. Aesthetic indications are slight asymmetries, ptotic breasts, large breasts where the reconstructed breast will be smaller than the original breast and where a Wise pattern is not indicated. Oncologic indications are risk-reducing mastectomies and tumors close to the NAC where resection would compromise the vitality of the NAC. We evaluated the healing of the NAC and the NAC position with regard to the breast shape. From 2019–2022, 296 autologous flaps were used for breast reconstruction. In 36 flaps, this technique was applied. Eighteen flaps were bilateral (nine patients). In total, we performed 15 inner thigh flaps and 21 DIEP flaps. No flap or NAC loss occurred. There was no wound healing complication at the breast, and no adjuvant chemotherapy or radiation therapy needed to be postponed. The advantages of this technique are (1) scar reduction with only one periareolar scar on the breast, which is also well concealed; (2) oncological safety in relation to the nipple and optimal visibility of the mastectomy cavity, which allows a meticulous mastectomy, especially important in risk-reducing mastectomies; (3) generally, fewer wound healing problems, especially fewer than with Wise pattern incision; (4) primary adjustment of the skin envelope and positioning of the NAC are easier to perform than in a secondary procedure; and (5) that the NAC is spared, and no secondary reconstruction is necessary. Disadvantages are that (1) the NAC must heal as a free graft and (2) that the sensitivity of the NAC is lower than in pedicled NAC transposition.

## 1. Introduction

Introduction of skin-sparing mastectomy (SSM) inevitably led to a paradigm shift in breast reconstruction [1]. Since then, primary reconstructions have increased and have become the therapy of choice. Direct-to-implant/expander reconstructions or immediate autologous reconstructions are oncologically safe and aesthetically pleasing [2].

Nipple-sparing mastectomy (NSM) was the next step in therapy [3]. Again, the data show high oncologic safety [4,5,6]. However, NSM makes achieving a good aesthetic result more complex because the mastectomy usually changes the footprint of the breast, but the position of the nipple–areola complex (NAC) and the skin envelope must fit [7]. This works best when mastectomy volume and volume of the reconstructed breast match and no radiation is applied [3,7].

This becomes more difficult in ptotic and/or large breasts. There, the skin envelope must be tightened and adjusted to the reconstructed breast volume, which is often smaller than the initial volume. In large breasts the incisions for a nipple sparing and skin reducing mastectomy (SRM) and safe outcome can collide [8,9].

Our preferred SSM incision since its popularization in 1998 is the periareolar/circumareolar with removal of the NAC [1]. By using this incision, adjustment of the skin envelope can be well accomplished in mildly-to-moderately ptotic breasts with a circumareolar resection [10]. Mastectomy skin can be safely dissected oncologically, and a shrinkage effect of the skin can be expected (Figure 1).

In the case of NSM and autologous reconstruction, skin adjustment via this incision is only The NAC itself is used as full thickness graft. However, in breast reconstruction, the technique of free NAC grafting has long been abandoned in favor of secondary nipple reconstruction and areola pigmentation, mainly because the position of the NAC is not always easy to determine, and a malposition which is difficult to correct can spoil the aesthetic result. After noticing that after circumareolar mastectomy, the NAC reconstruction is mostly positioned at the flap skin, we started to use the NAC as a free graft. We describe our technique consisting of the periareolar incision in mastectomy and following immediate autologous reconstruction and free NAC grafting and evaluated NAC healing and positioning.

## 2. Materials and Methods

In a retrospective analysis, we evaluated the technique of free NAC grafting in immediate autologous breast reconstruction. We have performed this technique since 2019. Patients in this study were operated on from 2019 to 2022. The patients were followed for 6 months to evaluate the healing and the NAC position on the breast. After that, the final positioning of the breast is usually completed. Aesthetic indications are slight asymmetries, ptotic breasts, large breasts where the reconstructed breast will be smaller than the original breast, and a Wise pattern is not indicated. Oncologic indications are risk-reducing mastectomies where patients want as little glandular tissue left as possible, and tumors that are close to the NAC such that resection would compromise the vitality of the NAC (Figure 2). Patients are always informed that nipple sensitivity is almost lost due to the free grafting.

Following surgical techniques were used depending on the anatomy:(1)Regular shape of the breast: If a scar-reducing NSM technique is to be used in a non-ptotic-shaped breast and the mastectomy and flap volume correspond to each other, a periareolar incision is made. The NAC is removed as a free graft, and retromamillary tissue is controlled by frozen section histology. If there is no tumor, the NAC is freely grafted. The nipple is hollowed out, and only minimal tissue from the lactiferous ducts is left in the nipple. After flap fixation, the patient is put into a sitting position intraoperatively, and the position of the NAC on the flap is determined by “tailor tacking”. If the flap is approximately the same size as the breast, the new NAC position is approximately the same as the old one. If the flap is smaller, the new position must be adjusted. Equally important as the distance between sternal notch and NAC upper edge is that between the NAC’s lower edge and the inframammary fold (IMF), which usually should not exceed 6–7 cm to provide lower pole control. With an areolar ring corresponding to the NAC, the new position is marked, and the flap dermis of the new NAC position is completely cut. The mastectomy skin is fixed to the new position (Figure 3). After that, the NAC is grafted to the flap dermis. The graft is secured with a tie-over or, if available, a negative-pressure wound therapy (NPWT) at 80–100 mm/HG.(2)Large nipples: In the case of a large nipples, a grafted NAC loses projection when well thinned. In this case, it can proceed as above, but additionally, a new nipple is reconstructed at the new NAC position from the flap’s dermis with a modified star-flap, and the NAC graft is put over the reconstructed nipple.(3)Ptotic/Large breasts: In ptotic breasts (minimal to moderate ptosis) or large breasts, the reconstruction results in less volume. The breast is measured preoperatively in a standing position. The new NAC position is marked. The upper edge should not be higher than 6–8 cm from the upper breast border. The distance between the lower edge of the NAC and the IMF should be equal to the volume of the flap. For a smaller breast, about 6–7 cm—for a larger one, 7–8 cm.

Vertical skin resection can be determined as in a mastopexy by guiding the breast sideways in relation to the midline of the breast in the IMF. The NAC is harvested after marking with an areolar ring. The outer circle is now used as a mastectomy incision (for example, Figure 4). The cranial and caudal positions of the mastectomy skin in the midline of the breast are marked with a stapler to avoid distortion. The procedure is then performed as described above. The excess skin is gathered to the new NAC position according to the markings.

## 3. Results

From 2019–2022, a total of 296 autologous flaps were used for breast reconstruction. In 36 immediate reconstructions (27 patients), this technique was applied (overview in Table 1); 18 flaps were bilateral (9 patients); 15 inner thigh flaps and 21 DIEP flaps were used for reconstruction (Figure 5, Figure 6, Figure 7, Figure 8 and Figure 9). No flap loss or NAC loss occurred (see Table 2). There was no wound healing complication or inflammation at the breast. Some minor hypopigmentations of the NAC occurred, which went unnoticed by the patients. One NAC showed a slight hyperpigmentation. All patients were satisfied with the resulting NAC, however one patient complained about the loss of projection of the grafted nipple, compared to the contralateral side. After 6 months, breast volume, skin envelope, and NAC positions were matching, and there was no distortion or malpostion of the NAC on the reconstructed breast. The NAC stayed in the position where it was initially placed. No adjuvant chemotherapy or radiation treatment had to be postponed. There were no further operations required for nipple corrections. No recurrence of cancer was noted in the follow-up period.

## 4. Discussion

We could show that in our cohort the circumareolar mastectomy in combination with autologous flaps and free NAC grafts yields good results. There was no impaired NAC healing. The new breast volume, skin envelope, and NAC positions matched. Furthermore, there was no malposition of the NAC after flap healing was completed.

Using the NAC as a free graft is an old technique that was applied for large breast reductions and early breast reconstructions [11,12,13]. In breast reconstruction, the technique has long been abandoned in favor of secondary nipple reconstruction and areola pigmentation. Among the reasons were that the position of the NAC is not always easy to determine in a primary reconstruction, and an NAC malposition on an otherwise well reconstructed breast can spoil the desired result. Additionally, oncological safety concerns were named to refrain from NAC grafting. Recently, however, the technique has experienced a renaissance in large breast reductions, gender-affirming surgery, and in the context of reconstructive breast surgery [14,15,16,17].

Only a few studies compared sensation of free NAC grafts to pedicled ones. In addition, the different pedicles come with different postoperative sensations. One study from 1974 evaluated the sensitivity, two-point discrimination, and erectility of free NAC grafts and controls (patients before surgery) in patients with breast reductions. Sensation was reduced to 65% in the free grafts in pin prick tests compared to non-operated patients. However, two-point discrimination was similar, and 72% of the free NAC grafts had regained erectility [18]. Another study compared free NAC grafts with inferior pedicle breast reductions. There sensation was similar in both groups for areola, but nipple sensation was better in the pedicled group. Erectile response was better in the pedicled group (90% vs 58% for free NAC grafts) [19]. Still, it is shown that even free NAC grafting can result in reactivation of erectile tissue within the NAC. However, grafting a NAC on a breast reduction will likely restore more sensitivity than grafting the NAC on a free flap after mastectomy.

Due to the fact that for oncologic reasons, the nipple in the NAC graft is thinned out as much as required in our surgery, a free NAC graft in a breast reconstruction will face some loss projection compared to a graft in an aesthetic procedure. Doren et al. describe that the nipples in their retrospective study maintained 59% of their initial projection [14]. There was very good healing in their group. However, sensitivity was not reported.

Incisions for NSM remain much discussed, in terms of (1) postoperative vitality of the nipple and mastectomy skin (2) leaving residual glandular tissue behind the nipple and in areas difficult to access from the incision; (3) visible, aesthetically poor scars; and (4) usability of existing scars for secondary surgery [3,4,5,6,7,8,9,20,21]. This situation affects younger patients and those undergoing prophylactic mastectomy more due to genetic alteration. Oncologically safe NSM requires subtle dissection that should leave as little glandular involvement as possible while sparing the blood supply to avoid necrosis [20,21].

In large and/or ptotic breasts, it is more challenging than in normal-shaped breasts, as the skin envelope must be adjusted to the new breast volume and the different footprint of the new breast using an SRM. When using the Wise pattern to reduce the skin envelope, very gentle dissection is vital in order to avoid necrosis at the mastectomy flap and NAC [9]. Nevertheless, wound healing problems and necrosis lead to additional scars and delays in the onset of adjuvant therapy [20]. Some authors therefore describe a staged procedure starting first with a breast reduction to adjust the skin envelope, which, however, can only be performed in risk-reducing mastectomies without cancer [22].

As described, we prefer the circumareolar incision for SSM in “normal”-shaped breasts with removal of the NAC. This incision has two important advantages: firstly, it provides a superior overview of the gland and allows the safe removal of all parts of the gland. One can easily access the axillary lobe and, if desired, the axilla. In a multicenter study, Papassotiropoulos et al. discovered overall residual breast tissue in 51% of the studied mastectomies [21]. The proportion increased in NSM with pedicled NAC, where residual tissue was present for almost 69%. This was confirmed in MRI studies, in which SSM had 13% residual tissue and NSM led to 50% of residual tissue. Secondly, if one replaces the skin of the NAC with the skin of the autologous tissue, one can preserve the shape of the breast. This leads to appropriate symmetry, especially in “normal”-shaped breasts, when the volumes of the mastectomy and the reconstruction correspond to each other.

In NSM with autologous tissue reconstruction, the pure circumareolar incision is conventionally not possible. Incisions are made in the inframammary fold (IMF), on the breast, semiperiareolar with vertical or horizontal extensions, or in the Wise pattern for large/ptotic breasts [23].

For small aesthetic mastopexies, the circumareolar (periareolar) incision has proven effective. A tightening of a few centimeters is thus very possible. This procedure is suitable for tubular breasts or for lifts that do not require wide cranialization of the nipple. Although some describe good results with this approach in larger mastopexies, the risk remains that the traction and tension around the areola will result in an enlarged areola and a hypertrophic periareolar scar [23].

In periareolar SSM, tightening of the skin envelope can be well accomplished via the circumareolar incision. In addition to the planned excision, there is a shrinkage effect of the skin. If the volume of the new breast is smaller than the initial volume or if there is excess skin, the skin envelope must be adjusted. Otherwise, the result of breast reconstruction will not be aesthetically pleasing.

In the case of NSM, skin adjustment via this circumareolar incision is only possible if the NAC is used a full thickness graft. This technique is well suited to removing excess skin and adjusting the skin to the new volume. It allows cranialization of the NAC if necessary, but also control of the distance between the nipple and the IMF, an important parameter. Excess skin would make the breast appear ptotic and flat. There is also a risk of seroma between skin and flap.

The prerequisite is to abandon the monitor island to assess the vitality of the flap and to de-epithelialize the site of the NAC after the flap inset. In our experience, after a periareolar mastectomy, one almost always plans the new nipple in the skin island of the reconstructed flap. The mastectomy thins the skin, which in turn shrinks without the need to widely resect it primarily. Moreover, if there is a large amount of skin and the flap volume is smaller than the original breast, a tension-free suture is given. Consecutively, enlargement of the areola and a hypertrophic scar can be avoided. Mastectomy, reconstructive procedure and mastopexy can be achieved with only a periareolar scar.

In our experience, the problem of wound healing disorders has been reduced compared to the Wise pattern incision. With this technique, the mastectomy flaps are safely perfused because of their wide bases.

In very large breasts that are reconstructed with smaller volume or severe ptotic breasts where the periareolar approach cannot be applied, we use an immediate Wise pattern. We use a caudal dermis flap, which would otherwise be resected, to improve breast shape and improve blood flow when using the Wise pattern. In cancer cases, we have experienced good results regarding pedicled NAC viability if a “delay” periareolar incision with dissection from the underlying gland is performed 5–7 days before reconstructive surgery [9]. However, a free NAC graft can also be applied in a Wise pattern SRM.

The advantages of the described technique are (1) reduction of overall scars, which are additionally hidden under the periareolar skin and allow an almost scarless mastopexy. The NAC stays together as a complex and retains its original color. (2) Oncological safety in relation to the nipple. Thinning the NAC as much as possible and leaving almost no residual retromamillary breast tissue leads to increased oncological safety. Additionally, optimal visibility of the mastectomy cavity allows a meticulous mastectomy, especially important in risk-reducing mastectomies, leaving less residual breast tissue. (3) Better hemostasis is possible and fewer wound healing problems arise, especially compared to Wise pattern incision. (4) Primary adjustment of the skin envelope and positioning of the NAC are easier to perform than secondary. (5) The NAC is spared, and no secondary reconstruction is necessary. Furthermore, keeping the NAC after the reconstructive operation is appreciated by the patients. Otherwise, nipple reconstruction is performed only after the completion of chemotherapy and radiation—on average with 3.2 procedures over 20.8 months [16].

Disadvantages are (1) slow healing of the NAC as a free graft. Healing always takes 10–14 days; sometimes, minor wound healing problems may occur. If healing is reduced there might be some depigmentation of the areola. However, these problems can also occur with a regular NSM or with a pedicled NAC if its position needs to be cranialized. (2) As a free graft, the sensitivity of the NAC is lower than that of pedicled NAC transposition. Unfortunately, there are no broad studies on the quality of sensitivity and possible preserved contractility depending on the different incisions and pedicled transposition that allow a comparison or describe a favored incision. (3) Depending on the extent of the coring of the nipple, it can lose projection. (4) By grafting the NAC onto the flap, flap skin monitoring is no longer possible. If desired, probe monitoring can be applied. However, if no flap skin is necessary and the flap is vital in the first 20–30 min after anastomosis, monitoring is not necessary in our opinion.

## 5. Conclusions

The free NAC graft is an option for the patients that are not eligible for regular NSM for technical, aesthetic, or oncological reasons. The free NAC graft and the circumareolar incision expand the possibilities of breast reconstruction and allow primary matching of skin and volume. Mastectomy, reconstructive procedure, and mastopexy leave behind just the periareolar scar. This is performed in a single procedure, and quality of NAC grafting appears better than in a secondary reconstruction. The patients welcome the fact that they are able to keep their nipple and areola.

## Figures and Tables

**Figure 1 jpm-12-01588-f001:**
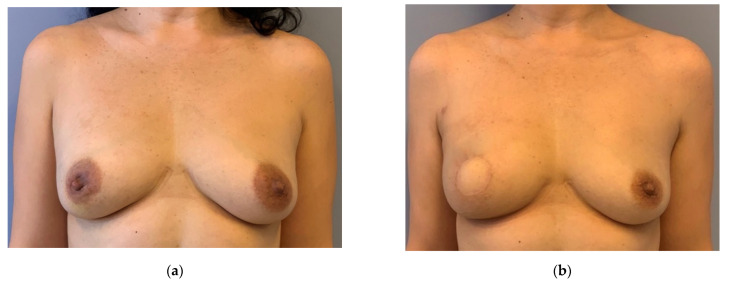
Example of circumareolar SSM. (**a**) Preoperative: 46 years old patient with invasive breast cancer on the right side. She underwent a circumareolar SSM and DIEP-flap reconstruction without NAC reconstruction. (**b**) Postoperative: 6 months after surgery and radiation therapy. The skin island replaces the areola. No other scars on the breast.

**Figure 2 jpm-12-01588-f002:**
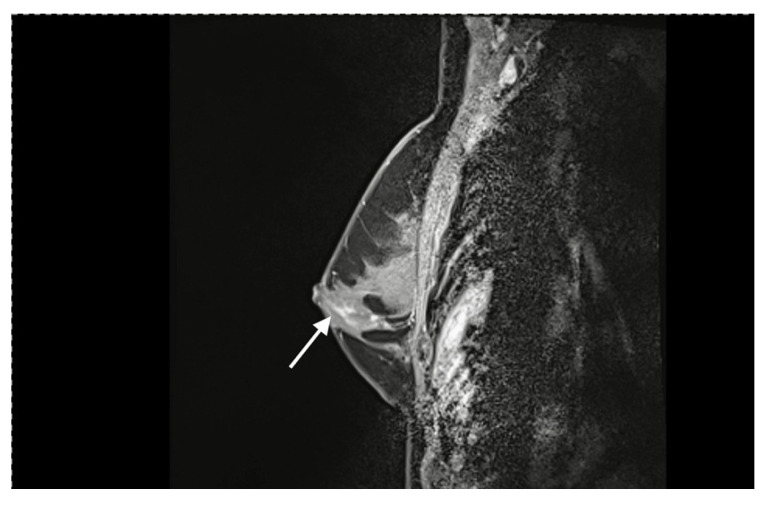
This MRI shows extensions of the cancer (arrow) in close proximity to the NAC. In these cases, the NAC can be used as free graft after retromamillary histology shows free margins.

**Figure 3 jpm-12-01588-f003:**
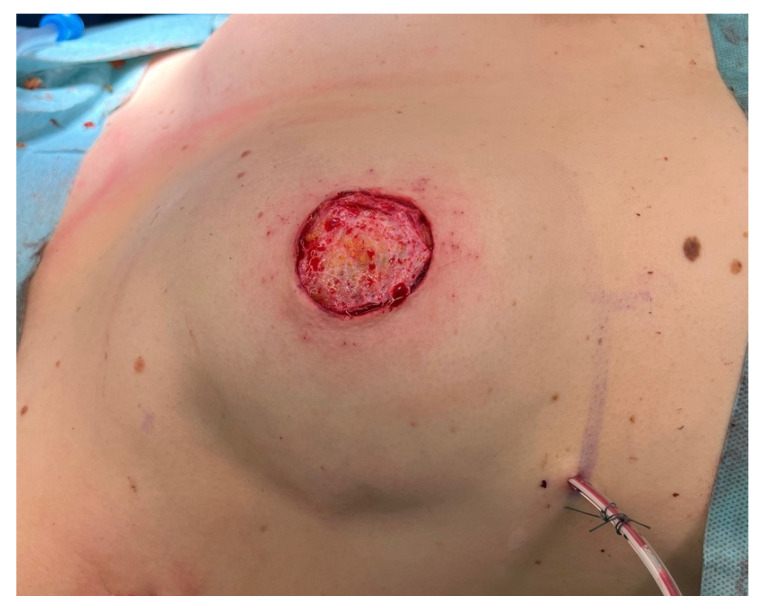
Mastectomy of the right breast (without ptosis) through a circumareolar incision. Inserted inner thigh flap through that incision. With an areolar ring that corresponds to the diameter of the previously harvested NAC, the position of the NAC is marked and then deepithelialized on the flap. The mastectomy flap is sutured to the dermis of the flap. The NAC is then grafted.

**Figure 4 jpm-12-01588-f004:**
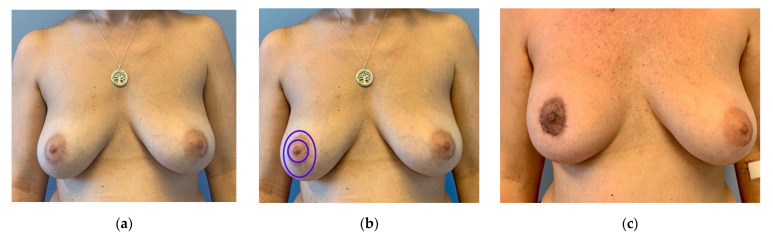
(**a**) Preoperative: 35-year-old patient with a multicentric invasive cancer on the right side required a mastectomy. Moderate ptosis. Right side slightly larger and more ptotic than left side. Sternal notch to nipple distance 24 cm, nipple to IMF distance 12 cm. (**b**) Schematic marking of the circumareolar incision to harvest the NAC (inner circle) and skin-reducing incision (outer circle). Without skin adjustment, the new breast shape would be flat. (**c**) Three months after circumareolar mastectomy, DIEP-flap reconstruction and free NAC graft (histologically controlled). Additional circumareolar skin was resected. Mastectomy and flap weight were identical (420 g). Patient is undergoing adjuvant chemotherapy. Slight hyperpigmentation of the NAC is visible. An alignment of the left breast is planned after completing adjuvant therapy.

**Figure 5 jpm-12-01588-f005:**
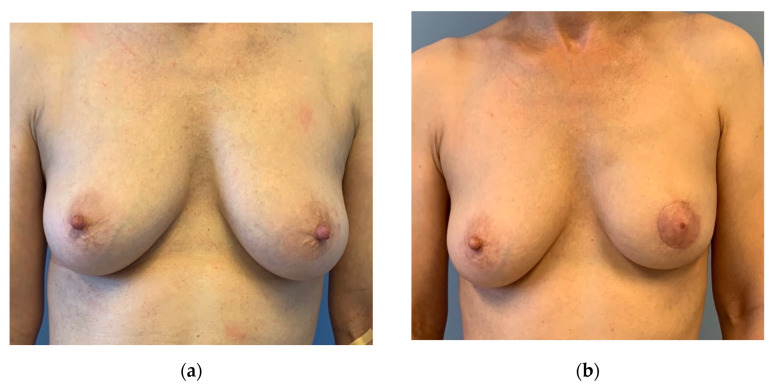
(**a**) Preoperative: 51-year-old patient with a multicentric invasive cancer on the left side and a large DCIS. Patient underwent neoadjuvant chemotherapy. Mild ptosis. (**b**) Three months after circumareolar mastectomy, inner thigh flap and free NAC graft (histologically controlled). Mastectomy weight and flap weight were almost identical. Circumareolar skin was resected at the lower pole to reduce the length of the nipple to IMF distance. An alignment of the right breast was not desired. No other scar at the breast apart from the periareaolar.

**Figure 6 jpm-12-01588-f006:**
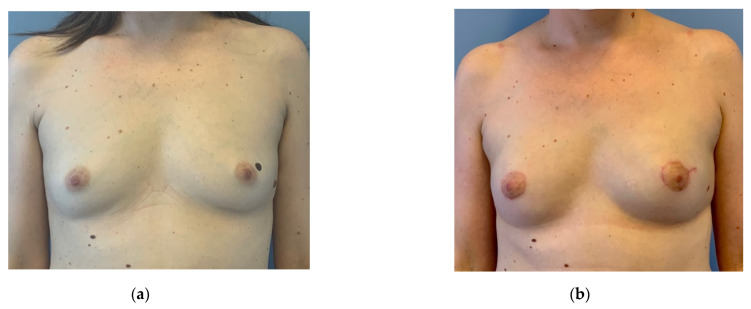
(**a**) Preoperative: 37-year-old patient with triple negative invasive breast cancer right, receiving neoadjuvant chemotherapy. No genetic mutation was detected. The patient desired a bilateral mastectomy. (**b**) Three months after bilateral circumareolar mastectomy and reconstruction with inner thigh flaps and free NAC grafts. No other scars evident at the breasts apart from the circumareolar. The nevus left was also excised.

**Figure 7 jpm-12-01588-f007:**
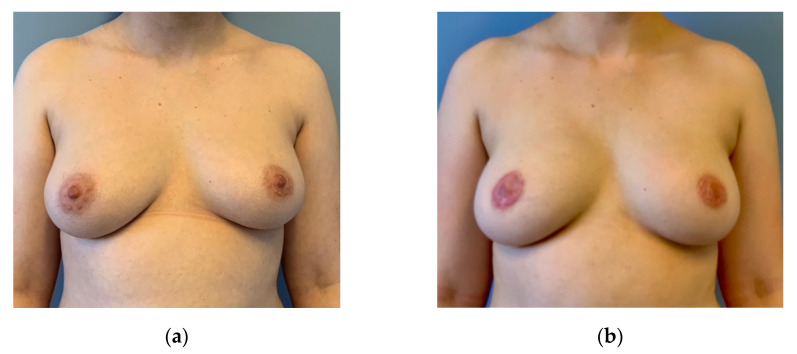
(**a**) Preoperative: 28-year-old patient with BRCA 1 mutation asking for risk-reducing mastectomy. Right side larger and more ptotic than left side. Sternal notch-to-nipple distance and nipple-to-IMF distance 2 cm longer than left side. (**b**) Three months after bilateral circumareolar mastectomy and reconstruction with bilateral DIEP-flaps and free NAC grafts. Flap weight was around 500 g on both sides. Mastectomy weight was 570 g on the right side and 500 g on the left side. The initial asymmetry was improved without additional scars.

**Figure 8 jpm-12-01588-f008:**
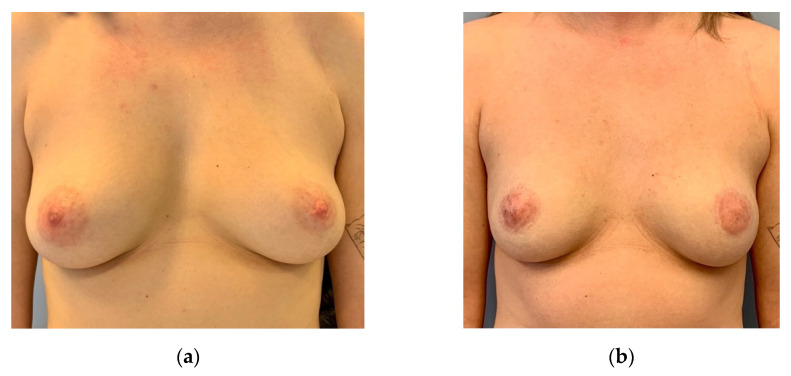
(**a**) Preoperative: 26-year-old patient with BRCA 1 mutation and slight asymmetry asking for risk-reducing mastectomy. (**b**) Five months after bilateral circumareolar mastectomy and reconstruction with bilateral inner thigh flaps and free NAC grafts. No other scars evident at the breasts apart from the circumareolar. Postoperative MRI reveals almost no residual glandular tissue.

**Figure 9 jpm-12-01588-f009:**
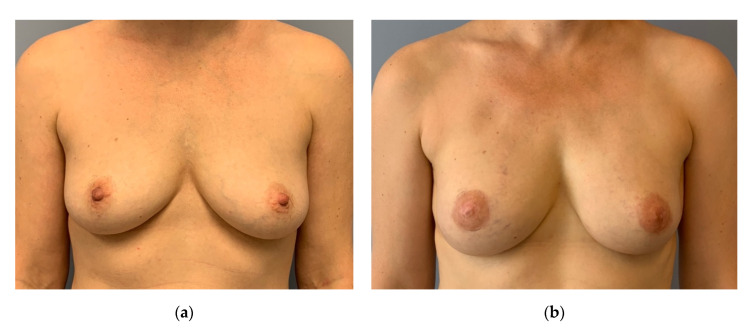
(**a**) Preoperative: 43-year-old patient with a mild ptosis and with BRCA 1 mutation receiving a risk-reducing mastectomy. (**b**) Four months after bilateral circumareolar mastectomy and reconstruction with DIEP flaps and free NAC grafts. No other scars evident at the breasts apart from the circumareolar.

**Table 1 jpm-12-01588-t001:** Data of the study group.

	Circumarealor Mastectomy, Autologous Flap and Free NAC Graft (*n* = 36 Flaps, 27 Patients)
Age at surgery	Mean 37 (24–56)
Side (uni/bilateral)	18 unilateral/18 bilateral (9 patients)
DIEP-flaps/inner thigh flaps	21/15
Cancer/preventive mastectomy	26/10
Regular breast shape/large nipple/ptotic breasts	14/2/20

**Table 2 jpm-12-01588-t002:** Results after mastectomy, autologous breast reconstruction and free NAC grafting.

	36 Reconstructions with Free NAC Graft
Flap loss	none
NAC loss	none
Wound healing complication, mastectomy flap necrosis	none
NAC malposition on breast	none
Loss of nipple projection	4
NAC hypopigmentation	5
NAC hyperpigmentation	1
Additional NAC surgery	none

## Data Availability

Supporting data are available from the authors upon request.

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
