# Peer review of "Autologous Breast Reconstruction with Free Nipple–Areola Graft after Circumareolar (Skin Reducing) Mastectomy"

_jpm, 2022, doi:10.3390/jpm12101588_

Round 1

Reviewer 1 Report

The abstract needs quantification. Follow the standard procedure for sample figure representation. The paper lags scientific analogy and statistical tests. The paper is a collection of data for a case study without any quantified approach or results. The conclusion needs modification. Total methodology has to be rearranged for a better understanding of the article.

Author Response

The abstract needs quantification. Follow the standard procedure for sample figure representation. The paper lags scientific analogy and statistical tests. The paper is a collection of data for a case study without any quantified approach or results. The conclusion needs modification. Total methodology has to be rearranged for a better understanding of the article.

Thank you for reviewing our manuscript.

Although the topic may appear simple, our paper covers several aspects: type of mastectomy, NAC healing and positioning on the breast, flap volume and skin envelope. All contribute to the aesthetic outcome. All therapies should also not delay adjuvant oncological therapies. Additionally, patients frequently ask for keeping their NAC, even if cancer is closely located. And, in risk reducing mastectomies patients often want the maximum safety, but also want to preserve the NAC.

These are “soft”, difficult to measure, parameters that in the end decide whether we perceive the reconstructed breast as aesthetically pleasing or not. Symmetry is another important point in reconstruction.  

We presented our surgical technique in order to improve the above-mentioned points and to show that using the NAC as a free graft leads to an aesthetically appealing outcome compared to not using it, which is current practice. We are aware that our paper is a description of a technique, and this technique adds to the surgical armamentarium and offers a select group of patients with breast cancer a better outcome.

We modified the abstract and the introduction to clarify our aim. The used surgical techniques were presented more concisely. Data of the patients were summarized in 2 tables. Additionally, the discussion was changed so that free NAC and the literature are discussed first followed by the discussion on the mastectomy and skin adjustment.

I hope the paper now has improved. Thank you again for your efforts.

Author Response

Line 75 – Please present the techniques followed in a better way for a better understanding. For example: 

“ Following techniques were used depending on the anatomy: 

(1) Regular shape of the breast. If a scar-reducing (…) “ 

We presented the techniques as suggested for a better understanding

Figure 3 – _What is the importance of the (a) there?

(a) was a mistake, now removed.

Line 116 – _I suggest the introduction of a table with information concerning the different techniques and the findings for better visualization of the outcomes. If possible a comparison with the standard procedure would be important.

Two tables were added, one to present the group data, one to present the results. However, a comparison to “standard” mastectomy/incision is very difficult to perform as the standard is very heterogeneous.

Line 120 – _And about inflammation?

There was no inflammation. Healing was good in all cases. We added that there was no inflammation. 

Line 187 – _I do not recommend initiating a statement with (2).

Was changed according to the suggestion.

Thank you for your meticulous review and the valuable suggestions. We very much appreciate your time and effort. 

Reviewer 3 Report

Dear Editor and Authors,

Thank you for the opportunity to review the manuscript entitled “Autologous breast reconstruction with free nipple-areola graft after circumareolar (skin reducing) mastectomy improves oncologic safety and allows an almost scarless mastopexy.” The authors described their technique of skin sparing mastectomy consisting of circumareolar incision in SSM, keeping the NAC as a free graft, and immediate autologous reconstruction and immediate free NAC grafting on the flap. They included this technique in 36 from 296 flaps and presented several advantages of this technique. The study is interesting well-designed and should be of interest of the Journal readers (Congratulations to the authors on their great surgical experience resulting in own modification of SSM improving aesthetic results!). I have only some minor/organizational remarks:

          -  the title is too definite

-       Abstract: please specify the aim of the study (e.g. to present….to assess aesthetic results, complication rate etc.), include information what was assessed in the study protocol.

-       Introduction: as above, please specify the aim (“to describe our technique consisting of… and evaluate “what”…)

-       Material and methods – please describe what was analyzed (patients satisfaction? aesthetics of NAC? Symmetry /volumetric?/, complications, recurrences? - who evaluated these aspects?  How? -on Likert scale?)

-       Results –  “in the follow up period…” – what was the follow up period? This should be specified in Material and methods (minimum and mean follow up period)

-       You could start discussion with summarizing your major findings and then discuss similarities with the available literature

-       Congratulations on perfect reconstructions shown on figures!!! 

Author Response

I have only some minor/organizational remarks:

 -  the title is too definite

We have changed the title to: “Autologous breast reconstruction with free nipple-areola graft after circumareolar (skin reducing) mastectomy”

-       Abstract: please specify the aim of the study (e.g. to present….to assess aesthetic results, complication rate etc.), include information what was assessed in the study protocol.

-       Introduction: as above, please specify the aim (“to describe our technique consisting of… and evaluate “what”…)

We have included the aim/goal of evaluation in the abstract and introduction.

-       Material and methods – please describe what was analyzed (patients satisfaction? aesthetics of NAC? Symmetry /volumetric?/, complications, recurrences? - who evaluated these aspects?  How? -on Likert scale?)

The focus was on healing of the NAC and the position. During the flap healing a migration of the flap and the NAC is possible. This was always postulated and used as argument to not immediately reconstruct the NAC. We have now seen, that there is no significant migration and the NAC stays on the breast where it was initially placed. In bilateral cases the symmetry was preserved. We think, that this is the most important finding. The patients were asked if they were satisfied, however this was not done by a Likert scale.

-       Results –  “in the follow up period…” – what was the follow up period? This should be specified in Material and methods (minimum and mean follow up period)

We clarified this.

-       You could start discussion with summarizing your major findings and then discuss similarities with the available literature

We followed your suggestions

-       Congratulations on perfect reconstructions shown on figures!!! 

Thank you very much for your appreciation.

Round 2

Reviewer 1 Report

All the corrections are included and the paper modified. Hence the article may be accepted.